# Lymphatic Tissue Bioengineering for the Treatment of Postsurgical Lymphedema

**DOI:** 10.3390/bioengineering9040162

**Published:** 2022-04-06

**Authors:** Cynthia J. Sung, Kshitij Gupta, Jin Wang, Alex K. Wong

**Affiliations:** 1Division of Plastic Surgery, City of Hope National Medical Center, Duarte, CA 91010, USA; cysung@coh.org (C.J.S.); ksgupta@coh.org (K.G.); jwang@coh.org (J.W.); 2Keck School of Medicine, University of Southern California, Los Angeles, CA 90033, USA

**Keywords:** lymphatic tissue bioengineering, lymphatic regeneration, lymphedema, lymphangiogenesis, stem cells, scaffolds

## Abstract

Lymphedema is characterized by progressive and chronic tissue swelling and inflammation from local accumulation of interstitial fluid due to lymphatic injury or dysfunction. It is a debilitating condition that significantly impacts a patient’s quality of life, and has limited treatment options. With better understanding of the molecular mechanisms and pathophysiology of lymphedema and advances in tissue engineering technologies, lymphatic tissue bioengineering and regeneration have emerged as a potential therapeutic option for postsurgical lymphedema. Various strategies involving stem cells, lymphangiogenic factors, bioengineered matrices and mechanical stimuli allow more precisely controlled regeneration of lymphatic tissue at the site of lymphedema without subjecting patients to complications or iatrogenic injuries associated with surgeries. This review provides an overview of current innovative approaches of lymphatic tissue bioengineering that represent a promising treatment option for postsurgical lymphedema.

## 1. Introduction

### 1.1. Lymphatic System

The lymphatic system is present throughout the body and is an important part of circulatory, immune, and metabolic systems. The lymphatic system serves three main functions: fluid homeostasis, immune regulation, and lipid absorption [1,2]. The physiologic function of the lymphatic system is to return extra interstitial fluid (ISF) to the blood circulation and maintain fluid homeostasis. ISF and extracellular matrix (ECM) are collectively called the interstitial space, or interstitium, and are localized outside the blood and lymphatic vessels [3]. Fluid, proteins, solutes, microorganisms, and leukocytes are extravasated from microcirculation into the interstitium and reabsorbed and cleared by lymphatic channels [4].

ISF enters the lymphatic system through blind-ended lymphatic capillaries in the periphery of the body and travels through lymphatic vessels before return to the blood circulation (Figure 1). Lymphatic capillaries consist of a single layer of lymphatic endothelial cells (LECs) with a discontinuous basement membrane, providing high permeability for macromolecules and cells. In addition, discontinuous basement membrane and a lack of mural cells allow LECs to have direct contact with ECM. The binding of LECs to ECM stimulates LEC proliferation, migration, and differentiation [2]. Lymphatic capillaries coalesce into precollector vessels and ultimately collector vessels. Collecting lymphatic vessels have characteristics distinct from lymphatic capillaries. They have a continuous basement membrane, contain semilunar-shaped valves and are covered by smooth muscle cells and pericytes that allow unidirectional flow [2]. Unilateral propulsion of the lymphatic fluid is maintained by intrinsic smooth muscle cell contraction, arterial pressure and surrounding skeletal muscle contraction. Collecting vessels merge, travel through the lymph nodes, and end in the subclavian vein where fluids are returned to the blood circulation.

The lymphatic system also plays a pivotal role in immune surveillance and lipid absorption. Collecting lymphatic vessels are connected to lymph nodes, serving as a conduit for foreign antigens as well as immune cells. Dendritic cells and T cells migrate from tissues to draining lymph nodes and return to blood circulation. As such, the lymphatic system is involved in immunologic responses, inflammation, cancer metastasis, and cardiovascular and metabolic disorders [4]. In the small intestine, lymphatic vessels reside in the intestinal villi and absorb dietary lipids and lipid-soluble vitamins in the form of chylomicrons from intestinal epithelial cells and return to the blood circulation [2].

### 1.2. Lymphangiogenesis

Lymphatic vessel development occurs during embryogenesis and in adults during inflammation and wound healing. During embryogenesis, lymphatic vessels arise from the cardiovascular system. The cardinal vein develops during early stages of embryogenesis by vasculogenesis. Vascular endothelial cells (VECs) that will become LECs acquire LYVE-1 expression. In LYVE-1^+^ cells, transcription factors, SOX18 and COUP-TFII, induce the expression of the critical lymphatic transcription regulator, PROX1. The expression of PROX1 upregulates VEGFR-3 and renders LECs more responsive to VEGF-C signaling, enhancing the LEC polarization and formation of lymph sacs lateral to the cardinal veins [5]. The lymph sacs give rise to primordial lymphatic structures and eventually the entire lymphatic system. During this process, VEGFR-3 and VEGF-C play a vital role in lymphatic sprouting, migration, and maintenance [2]. LECs are committed to the lymphatic lineage and mature into lymphatic vessels through centrifugal growth from the cardinal vein [5]. Podoplnin is also expressed by LECs during this stage which contributes to platelet aggregation, blocking lymphatico-venous connections and separating the lymphatic system from the venous system. Lymphatic structures further differentiate and develop into capillaries and collecting vessels.

## 2. Lymphedema and Lymphatic Tissue Bioengineering

Lymphedema is defined as the excessive accumulation of protein-rich fluid in the extracellular space due to homeostatic disturbance of lymphatic system. It is one of the most prevalent lymphatic associated diseases that affect more than 250 million people worldwide. Lymphedema can be classified into primary and secondary causes. Primary lymphedema occurs from genetic abnormalities and developmental failure of lymphatics. It is relatively rare, affecting 1 in 100,000 individuals in the United States. Secondary lymphedema occurs from other disease processes, mostly commonly filariasis or injury associated with cancer treatments such as lymphatic tissue removal or radiation therapy, and affects 1 in 1000 people in the United States [1]. It is especially prevalent in the oncologic population with one in five women with breast cancer developing lymphedema after undergoing mastectomy [2]. Fluid stasis causes accumulation of macromolecular proteins and hyaluronan, which in turn attracts more water due to an increase in tissue colloid osmotic pressure [6]. Chronic lymphatic stasis also affects cellular behaviors and compositions in their microenvironment. The number of fibroblasts, keratinocytes, adipocytes, and inflammatory cells accumulates, and the tissue undergoes progressive pathologic changes, including chronic inflammation, fibrosis, and adipose tissue deposition.

Lymphatic accumulation can cause physical discomfort and pain due to swelling of soft tissues and immunologic complications due to dysregulation of immune cell trafficking. It significantly reduces every aspect of a patient’s quality of life. Lymphedema is a life-long condition that does not have effective curative treatments, placing a large burden on patients, society, and the healthcare system. With a rising life expectancy and increasing cancer survival, an increased number of people will be affected by lymphedema and associated complications.

Surgical strategies for lymphedema management are aimed at restoring physiologic function include lymphatic bypass and vascularized lymph node transfer (VLNT). These techniques are intended to create new channels and promote the drainage of lymphatic fluid through anastomosis with the venous system or incorporation of a functional lymphatic structure [7]. While studies report a long-term clinical improvement of lymphedema after surgical interventions, they are not without drawbacks; lymphatic transfer requires a patent lymphatic vessel for anastomosis, VLNT poses significant donor site complications, excision has a high risk of surgical site complications [7]. Surgical interventions can be unsuccessful in a subset of patients, especially those with advanced stage lymphedema, in which supportive care is the only remaining option. As lymphedema progresses to an advanced stage, severe and irreversible tissue changes occur such as fat hypertrophy and fibrosis. The fibrosclerotic alterations of the lymphatic tissues hinder the effective repair by existing techniques [8].

With the growing knowledge in lymphatic biology and tissue engineering and nanotechnology, there has been growing interest and demand for tissue engineering for the treatment of postsurgical lymphedema. Engineered lymphatic vessels or lymph nodes can be transplanted to repair diseased or defective lymphatic tissues. Moreover, a 3D construct of lymphatic structures will be a valuable tool for pathophysiologic and therapeutic studies of lymphedema. The engineered tissue construct can provide a disease model for in vitro experiments to elucidate poorly understood pathways or serve as a tool to assess safety and efficacy of drug therapies [9]. The current armamentarium of lymphatic tissue bioengineering includes stem cells, pro-lymphangiogenic factors, bioengineered scaffolds/matrices and mechanical stimuli. In this review, we aim to provide various approaches of lymphatic tissue bioengineering and regeneration.

## 3. Stem Cells

Stem cells are characterized by their capacity for self-renewal and multilineage differentiation. Based on their differentiation potential, stem cells are divided into totipotent, pluripotent, multipotent, oligopotent, and unipotent stem cells [10]. Totipotent stem cells refer to zygote and blastomeres which can form all the cell types in a body and the placenta. The pluripotent stem cells include embryonic stem cells (ESC) and induced pluripotent stem cells (iPSCs) that can give rise to all the cell types that make up the body. Multipotent stem cells are found in most tissues and differentiate into cells from a single germ layer. Mesenchymal stem cells (MSCs) are the most recognized multipotent stem cells that can be derived from a variety of tissue including bone marrow, adipose tissue, bone, Wharton’s jelly, umbilical cord blood, and peripheral blood. These cells tend to turn into mesoderm-derived tissue such as adipose tissue, bone, cartilage, and muscle. Oligopotent stem cells can self-renew and form two or more lineages within a specific tissue. Hematopoietic stem cells are a typical example of oligopotent stem cells, as they can differentiate into both myeloid and lymphoid lineages. Unipotent stem cells can self-renew and differentiate into only one specific cell type and form a single lineage. The purpose of this part is to summarize stem cell based lymphatic tissue bioengineering.

### 3.1. Embryonic Stem Cells (ESCs)

Embryonic stem cells are obtained from the inner cell mass of the blastocyst that is three to five days old. Embryonic stem cells can give rise to every cell type in the fully formed body, but not the placenta and umbilical cord. Lymphatic endothelial cells were successfully differentiated from VEGFR-2^+^ embryonic stem cells at day 3 on OP9 stromal cells. These endothelial cells were defined by the expression of LEC-specific markers, Prox1, VEGFR-3, Podoplanin, and LYVE-1 [11]. Kono et al. used 2 successive steps for LEC differentiation from ESCs. VEGFR-2^+^ E-cad^−^ mesodermal cells were first generated and enriched from ESCs. ESC-derived purified VEGFR-2^+^ cells were then cocultured with OP9 Cells onto type IV collagen-coated dishes in the presence of VEGF-C and Ang-1. However, the molecular basis of LEC induction activity by OP9 cells remains to be elucidated. The addition of VEGF-A and VEGF-C to murine ESCs and LECs led to the formation of embryoid bodies (EBs) at day 18. ESCs are collected from early-stage embryos, which only consists of 100–200 cells. The scarce source of human oocytes combined time-consuming processes greatly limits their widespread use. The ethical debate about using embryos needs to be considered as well. Moreover, as embryonic cells are allogenic, they pose an immunologic barrier to the patient.

### 3.2. Human Induced Pluripotent Stem Cells (hiPSCs)

Induced pluripotent stem cells (iPSCs) share similar characteristics with ESCs and can differentiate into LECs [12]. iPSCs are usually made from skin or blood cells. Researchers described an effective protocol to convert hiPSCs into cells similar to cord-blood endothelial colony-forming cells (CB-ECFCs), vascular endothelial cells and smooth muscle cells [9,10]. Studies showed therapeutic efficacy of hiPSC-derived endothelial cells in pre-clinical models of cardiovascular disease [11,12]. The hiPSC-ECs generated from the differentiation of hiPSCs using VEGF-A and bone morphogenetic protein-4 were purified based on positive expression of CD31. Rufaihah et al. demonstrated that FACS purification of CD31+ hiPSC-ECs produced a diverse population of ECs. Given heterogeneity of hiPSC-derived endothelial cells, authors induced differentiation of hiPSC-derived endothelial cells to lymphatic phenotype using VEGF-C and angiopoietin-1 [13]. VEGF-C, which acts on the VEGFR3 receptor, promotes migration of lymphatic endothelial progenitors from the vein. Ang-1 acts through the receptor tyrosine kinase, Tie-2 in regulating lymphatic vessel formation, sprouting, and lymphatic endothelial proliferation. High VEGF-A and VEGF-C concentrations with supplementation of Ang-1 in the medium promote the specification to lymphatic CD31+ hiPSC-ECs. Lee et al. found that an OP9-assisted culture system reinforced by the addition of VEGF-A, VEGF-C, and EGF most efficiently generated LECs, which were then isolated via FACS-sorting with LYVE-1 and Podoplanin [14]. These hiPSC-derived LYVE-1 + Podoplanin + cells exhibited a pure committed LEC phenotype, formed new lymphatic vessels, and expressed lymphangiogenic factors at high levels. These hiPSC-derived LECs improve wound healing by promoting lymphatic neovascularization through both lymphvasculogenesis and lymphangiogenesis. However, since xeno-free and well-defined culture conditions are ideal for clinical application, this approach of LEC differentiation requiring co-culture with mouse OP9 stromal cells may not be suitable [7,14].

### 3.3. Multipotent Adult Progenitor Cells (MAPCs)

Multipotent Adult Progenitor Cells (MAPC) are non-hematopoietic cells found in bone marrow stroma, which play a role in the maintenance of the hematopoietic stem cell niche. While these cells meet the International Society for Cell Therapy (ISCT) criteria for MSCs, they were perceived to be a more biologically primitive population than classical MSCs and had greater differentiation potential [13]. Bone marrow derived multipotent adult progenitor cells (MAPCs) have multi-lineage differentiation potential. Beerens et al. reported that, in vitro, MAPCs showed potential to differentiate down the lymphatic endothelial lineage. Culturing MAPCs with VEGF-A results in a heterogeneous mixture of arterial, venous, and lymphatic endothelial cells [15]. Unlike hiPSCs, exposure to VEGF-C did not increase lymphatic differentiation. In addition, MAPCs promoted lymphangiogenesis in vivo and restored lymph drainage across skin flaps by stimulating capillary and pre-collector vessel regeneration. Considering the proangiogenic effect of MAPC may affect tumor growth or metastasis, the proangiogenic response should be sufficiently balanced and locally controlled.

### 3.4. Adipose Tissue-Derived Stem Cells (ADSCs)

Adipose tissue-derived stem cells (ADSCs) are multipotent stem cells within adipose tissue and a promising source for lymphangiogenesis [14]. ADSCs have two lymphangiogenic mechanisms: paracrine secretion and direct differentiation into lymphatic endothelial cells [16]. Studies have shown LEC differentiation from human ADSCs by lentiviral Prox1 overexpression as well as IL-7 through AKT pathways [15,17]. In vivo studies on mouse lymphedema models demonstrated that ADSC injection resulted in greater lymphatic capillary density, tissue expression of VEGF-C, plasma levels of VEGF-C, and higher recovery from lymphedema [16]. When ADSC injection was combined with vascularized lymph node transplant, ADSCs promoted an increase in lymphatic vessels and preservation of transplanted lymph nodes [18]. ADSCs have been used clinically. A single injection of ADSCs into the axillary region together with a scar releasing fat graft procedure improved patient-reported outcomes and reduced use of conservative management in 10 BCRL patients at a 1-year follow-up. However, no significant changes in volume and quantitative lymphoscintigraphy on the lymphedema affected arms were noted at one year follow-up [19]. Nonetheless, as abundant and autologous sources of stem cells, ADSCs show the potential to be used in lymphatic tissue engineering and regeneration. Further studies are needed to establish standardized protocols and explore long-term outcomes.

### 3.5. Hematopoietic Stem Cells (HSCs)

Hematopoietic stem cells (HSCs) are a type of oligopotent stem cell that differentiate into cells of the blood and immune system. Adult mouse bone marrow derived HSCs give rise to functional vascular endothelial cells that express CD31, produce von Willebrand factor, and take up low density lipoprotein [13,14,16,20]. Jiang et al. showed that adult hematopoietic stem cells can give rise to LECs that integrate into lymphatic vessels of normal tissues and in newly formed tumors [8]. It is not well understood whether HSC-derived LECs potentiate lymphangiogenesis in tumors and therefore will need further studies.

### 3.6. Endothelial Colony-Forming Cells (ECFCs)

Endothelial colony-forming cells (ECFCs) were isolated from adult human peripheral blood and noted to express blood vascular or lymphatic-specific [21]. Even though a majority of ECFCs expressed high levels of VEGFR-1, two of the clones (ECFC L1 and L2) expressed very low levels of VEGFR-1 but high levels of VEGFR-3, podoplanin, LYVE-1, and prox-1. The lymphatic ECFCs responded to VEGF-C and VEGF-A while the blood ECFCs only responded to VEGF-A. Lymphatic ECFCs expressed higher levels of Wnt 5a and Wnt 5b, Notch 3 and Jagged 1, suggesting that Notch signaling during development of endothelial precursors could be playing a role in their lymphatic specification. This study demonstrates the presence of lymphatic precursor endothelial cells in adult peripheral blood that can be used for lymphangiogenesis.

Growing evidence has shown the regenerative potential of stem cells in the tissue bioengineering, and a variety of stem cells including embryonic, induced pluripotent, mesenchymal, and others, have been studied. Despite possessing great pluripotency and proliferative capacity, several challenges in altering the state of cells limit their widespread clinical translation of pluripotent stem cells [22]. Efficient differentiation of stem cells and reliable maintenance of the viability and potency of differentiated cells during the process are critical obstacles that need to be addressed [23]. Another concern associated with stem cells is safety. Inducing differentiation of stem cells into a lymphatic endothelial cell type with 100% efficiency is difficult and a small fraction of undifferentiated cells may remain, which can lead to neoplastic development [22]. iPSCs and adult progenitor cells overcome ethical and immunological barriers of embryonic stem cells and provide a more stable tissue source. Given the ease of availability and abundance, they represent an attractive option for lymphatic tissue engineering [14].

The limitations of stem cells can be overcome with bioengineering technology. Pro-lymphangiogenic factors can guide the fate of stem cells and their regenerative potential. Biomaterials provide a supportive microenvironment for cells to proliferate and differentiate. They can be engineered with a specific mechanical and biochemical property to improve proliferation and differentiation capacity and viability, and to preserve stem cell function. The engineering strategies to overcome these limitations and advanced clinical translation of stem cells are discussed in the following sections.

## 4. Prolymphangiogenic Factors

### 4.1. VEGF-C

Vascular endothelial growth factors (VEGFs) are known to play essential roles in the growth of both blood vascular and lymphatic endothelial cells. The VEGF family comprises five secreted glycoproteins including, VEGF-A, PIGF, VEGF-B, VEGF-C and VEGF-D that bind with differing specificities to the three VEGF receptors (VEGFR-1, VEGFR-2 and VEGFR-3) [24]. VEGFR-1 and VEGFR-2 are found chiefly in blood vascular endothelial cells while VEGFR-3 is expressed in lymphatic endothelia. Notably, exposure of endothelial cells to VEGF has been shown to induce survival via regulating phosphatidylinositol 3′-kinase/Akt signal transduction pathway [25]. A study by Benjamin LE et al. employing a switchable VEGF expression system in vivo has shown that VEGF functions as a survival factor for newly formed vessels while its withdrawal result in regression of preformed tumor vessels [26]. Notably, apart from the effects on survival and proliferation, supplementation of LEC with VEGF-A and VEGF-C in vitro has been demonstrated to promote the formation of lymphatic capillaries and it also led to the enhancement of the density of branching lymphatic capillaries, when compared to the effect observed in LEC cultured in factor-free control medium [27]. Employing a mouse model of adult lymphangiogenesis in regenerating skin, Rutkowski et al. demonstrated that the expression of VEGF-C amplifies in the initiation of lymphangiogenesis while it gradually decreases during the organization and maturation stages [28]. Likewise, lymphatic regeneration in nude mice following skin grafting materializes as a result of ingrowth of lymphatic vessels and re-assembly of prevailing lymphatics and this was found to be temporally and spatially associated with the VEGF-C expression pattern [29]. In another study, VEGF-C overexpressing transgenic mice were found to possess hyperplastic lymphatics, but it did not promote endothelial proliferation and vessel enlargement [30]. Likewise, VEGF supplementation in stereotypically regenerating skin led to hyperplastic lymphatic vessels; however, it does not augment the rate of LEC migration or functional lymphangiogenesis [31]. Adenoviral-mediated overexpression of VEGF-C in the skin and respiratory tract of nude mice led to blood vessel enlargement, tortuosity, and leakiness; however, its angiogenic effectiveness in sprouting new lymphatic vessels was very much abridged [32]. It has also been indicated that though exogenous supra-physiological doses of VEGF-C may induce lymphanagiogenesis, they, however, are also associated with inflammatory response and the newly formed vessels may be immature, leaky, and non-functional [33,34]. Thus, under physiological conditions, VEGF-C alone maybe insufficient to promote long-term lymphangiogenesis and would require additional input or mediators [35] as discussed in the following section. To address this, Kataru et al. have moderated the VEGF-C signaling by conditionally knocking out PTEN using the Vegfr3 promoter in LECs [36]. A plethora of studies on the other hand, have highlighted the role of VEGF-C in inducing excessive tumor lymphangiogenesis, increasing vascular permeability, which may also result in the higher rate of metastasis [32,37,38,39,40,41,42,43].

### 4.2. TGF-β

The transforming growth factor-β, long been regarded as a crucial growth factor in the pathogenesis of fibrosis [44], has recently been studied for its effect on lymphangiogenesis. TGF-β signaling has been shown to inhibit the proliferation, cord formation and migration of human dermal lymphatic microvascular endothelial cells (HDLECs) toward VEGF-C. The inhibitory effect was further reflected in the reduced expression of LEC-related genes, Prox1 and LYVE-1 in LEC cells [45]. In another study, the inhibition of TGF-β1 expression during wound healing has been demonstrated to accelerate lymphatic regeneration and LEC accumulation; at the same time, it caused impaired myofibroblast proliferation and lymphatic fibrosis [46,47]. These studies thus acclaim for a direct inhibitory role of TGF-β on lymphangiogenesis [45,46]. In contrast, TGF-β1 mediated upregulation of VEGF-C expression has been reported in renal tubular epithelial cells, collecting duct cells, and macrophages but not in fibroblasts, which in turn lead to lymphangiogenesis [48]. This effect was reversed by treatment with TGF-β type I receptor inhibitor LY364947. Likewise, the occurrence of sine oculis homeobox homolog 1 (SIX1) overexpression in cervical tumor cells induces the activation of the TGF-β-SMAD pathway, which in turn mediates VEGF-C augmentation leading to the promoting effects on migration of tumor cells, tube formation of lymphatic endothelial cells (LEC) in vitro and lymphangiogenesis in vivo [49]. Additionally, TGF-β treatment of bone marrow-derived macrophages led to the enhanced production of hyaluronan (HA) in which in turn stimulated VEGF-C expression and hence, enhanced lymphangiogenesis in a renal fibrosis model [50]. Work from the laboratory of Takei et al. have projected that TGF-β augments VEGF-C expression in several cell types including tubular epithelial cells, macrophages, and mesothelial cells, thereby promoting lymphangiogenesis in renal and peritoneal fibrosis [51].

### 4.3. FGF

The FGF family is comprised of ~23 polypeptide growth factors that modulate several cellular functions and physiological processes including migration, proliferation, differentiation, angiogenesis, wound healing, and tumorigenesis [52,53,54]. Exogenous FGF-2 supplementation promoted the cell proliferation and migration of newborn bovine lymphatic endothelial cells [55,56]. Notably, FGF-2 mediated lymphangiogenesis in the cornea displays a correlation with the up-regulated expression of MMP-2 and -9 [57]. More importantly, it was independent of the requirement of a preexisting vascular bed (angiogenesis) [57]. Recent studies have shed light on the existence of a collaborative interaction between VEGF-C and FGF-2 in promoting lymphangiogenesis in the tumor microenvironment which eventually favors lymph-node metastasis [58]. In concordance with this, FGF-2 mediated lymphangiogenesis in mouse cornea displayed susceptibility to anti-VEGFR-3 antibodies treatment. In yet another study, exogenous FGF-2 treatment led to the up-regulation of VEGF-C expression in vascular endothelial and perivascular cells [59]; while the suppression of endothelial FGF signaling led to the waning of VEGF signaling, decreased Erk1/2 and Akt phosphorylation in vitro and in vivo [60]. In a landmark study, Shin et al. identified FGFR-3 as a target gene for the transcription factor Prox1, [61] which is an essential component in the development of lymphatic vasculature. In addition, FGFR-3 inhibition either by siRNA or a tyrosine kinase inhibitor overturns the FGF2-induced LEC responses related to cell proliferation, migration and survival. Of note, LYVE1, a common lymphatic marker, was found to bind FGF2 and stimulate FGF2-induced signaling activation and lymphangiogenesis [62]. Nevertheless, it remains unclear if FGF2 directly induces lymphangiogenesis or exerts effects indirectly via VEGFC/VEGFR3 signaling or FGFR1 and VEGFR3 bind a common signal transducer in LECs, and thus needs more conclusive evidence and investigation.

### 4.4. PDGF

Platelet-derived growth factors (PDGFs) and their receptors (PDGFRs) are critical moderators of blood vasculature development. The PDGF family consists of dimers of two gene products; PDGF-AA, -AB, -BB, -CC and -DD; with their receptors being expressed on the blood vessels’ endothelial cells, indicating a direct role in angiogenesis [63]. The angiogenic activity of PDGFs has been shown to escalate in the presence of other angiogenic growth factors. For example, synergism between VEGFR2 and VEGFR3 in lymphatic endothelial cells led to significant upregulation of PDGF-B expression. Furthermore, VEGFR3 signaling served as a critical determinant of PDGF-BB-dependent capillary stabilization in lymphatic endothelial cell [64]. Several studies have highlighted the role of the PDGF/PDGFR axis in the lymphatic spread of tumor cells and tumor lymphangiogenesis and there is compelling evidence, indicating that PDGF-BB acts as a direct lymphangiogenic factor [65,66,67,68,69,70]. In addition, PDGF family growth factors have also been shown to exert a stimulatory effect on the VEGF family members. For example, cholangiocarcinoma cells recruit and activate cancer-associated fibroblasts (CAFs) by secreting PDGF-D. Fibroblasts stimulated by PDGF-D, in turn, led to enhanced production of VEGF-C/A, increased LEC recruitment and 3D assembly, increased LEC monolayer permeability, and transendothelial EGI-1 migration [71]. A study from Cao et al. has shown that PDGF-BB stimulates MAP kinase activity and induces lymphatic endothelial cells’ motility in vitro; while PDGF-B expression in transplanted primary murine sarcomas induced lymphangiogenesis and lymph node metastasis in mice [65].

### 4.5. Retinoic Acid

Various pro-lymphangiogenic growth factors have either shown unwanted secondary effects on non-lymphatic tissues or have demonstrated proliferative effects on cancer cells, thereby limiting their use in clinical setting [72,73]. To circumvent these issues, we had screened several non-growth factor compounds and showed the essential role of retinoic acids, including 9-cis Retinc acid (9-cis RA) and all-trans Retinoic acid (ATRA) and RAR agonists (TTNBP and AM580) in promoting the lymphatic endothelial cell proliferation in vitro [74]. Retinoic Acids (RAs) are metabolites of vitamin A known to regulate various biological processes via modulating genes relevant for cell proliferation, differentiation, apoptosis, and metabolism. Through comprehensive in vitro and in vivo studies, we have demonstrated that 9-cis RA augment lymphatic endothelial cell proliferation, migration and vessel regeneration via the activation of FGF receptor signaling [74]. Moreover, 9-cis RA activates cell cycle progression by modulating the expression of CDK inhibitors and aurora kinases through an Akt-mediated genomic and Prox1-mediated non-genomic actions. 9-cis RA was also found to enhance in vivo lymphangiogenesis as evident from the mouse trachea, Matrigel plug and cornea pocket assays. Additionally, intraperitoneal administration of 9-cis was found to increase the lymphatic drainage and lymphangiogenesis in a rodent model of secondary lymphedema [75]. Another landmark study from our laboratory demonstrated that sustained delivery of 9-cis RA contained within an implantable, single-use depot drug delivery system, when placed within the surgical site intraoperatively, significantly reduced lymphedema, which accompanied faster lymphatic clearance and increased lymphatic density [76]. Overall, these findings advocate for the therapeutic relevance and applicability of 9-cis RA for the treatment of postsurgical lymphedema in humans. Ongoing work in our laboratory includes investigation 9-cis RA and other retinoic acid family members with the ultimate goal of understanding the precise molecular pharmacology of retinoic acid associated lymphangiogenesis. 

### 4.6. Other Factors

In addition to the role of VEGF, FGF and PDGF family members, emerging studies have described the role of other biochemical stimuli/molecular signals in promoting lymphangiogenesis.

Members of the angiopoietin family (Ang-1 and Ang-2), a ligand for the Tie-2 receptor tyrosine kinase, have been shown to modulate communications between endothelial and periendothelial cells [77]. Growth factors (VEGF and FGF) supplementation in the bovine microvascular endothelial cells was reported to escalate the mRNA expression of Ang-2, thus modulating their angiogenic activity [78]. Additionally, adeno associated virus-mediated overexpression of Ang-1 in mice not only promote the proliferation and enlargement of lymphatic vessel, but also led to the sprouting of new vessels and cutaneous hyperplasia; these effects though worked in synergy with VEGFR-3 up-regulation [77]. In another study, mice lacking Ang-2 were shown to display lymphatic vessel defects, indicating the possible requirement of Ang-2 in the vasculature development and stabilization [79]. However, the precise role of Ang-2 in pathological lymphangiogenesis remains less understood. In wound healing, Ang-2 overexpression induces lymphangiogenesis at the wound margin [80]; however, it remains unclear if this is a direct effect of Ang-2 on lymphatic vessels, since Ang-2 also boosts the influx of inflammatory cells [81,82], which stimulate lymphangiogenesis by the secretion of growth factors and cytokines (TNF a).

Likewise, HIF-1a has been proposed as a central regulator of lymphangiogenesis [83,84] blockade of HIF-1a increases edema and lymphatic fluid stasis following wounding. Using mouse models of lymphatic regeneration, Zampell et al. have shown that both hypoxia and gradients of lymphatic fluid stasis stabilizes HIF-1α expression, which in turn alters the expression of lymphangiogenic cytokines, including VEGF-C, to coordinate lymphangiogenesis in the process of wound healing [85].

Adrenomedullin (AM), a vasoactive peptide, has been implicated in angiogenesis and lymphangiogenesis during embryonic development, wound healing, and cancer [86]. AM haploinsufficiency predisposes mice to secondary lymphedema following hind limb injury [87]; this effect was, however, restored following systemic injection of AM. Very recently, AM was shown to drive cardiac lymphangiogenesis following injury in a surgical model of myocardial infarction [88].

## 5. Scaffolds

A critical element in tissue engineering is a scaffold that can facilitate cell growth, organization, and differentiation [9]. Scaffolds deliver cells and growth factors to the desired location in the body and act as analogues to the extracellular matrices naturally found in the tissue [89]. The important consideration for using scaffolds is the viability and growth of seed cells after their attachment to the scaffold. Scaffolds also control growth factor binding and transport, which can be tailored through matrix composition and nano- and microporosity [9]. An ideal scaffold should have the following characteristics: appropriate architectures that provide physiologic and mechanical properties, good biocompatibility to the existing microenvironment, and low immunogenicity and inflammation. Various sources of scaffolds have been described for lymphatic tissue engineering including hydrogels, synthetic or natural matrices, cellular sheets, decellularization of existing tissues, and 3D bioprinting (Table 1).

### 5.1. Hydrogels

Hydrogels are highly hydrated networks of crosslinked polymer chains from natural or synthetic origins. They have been extensively used in both clinical and research areas for cell manipulations, drug delivery, tissue engineering, and 3D bioprinting [94]. Hydrogels have structural similarities to extracellular matrices found in the body and are considered biocompatible [89]. Hydrogels have been used to reconstruct lymphatic system and lymph nodes [94]. A wide range of natural and synthetic polymers have been studied for this application. Natural polymers used for hydrogels include collagen and gelatin, fibrin, hyaluronate, alginate, and agarose.

Fibrin and collagen hydrogels were used to create dermo-epidermal skin grafts that contained lymphatic and blood capillaries by combining human dermal microvascular endothelial cells (hDMECs) and fibroblasts in fibrin or collagen hydrogels [27]. When the skin grafts were transplanted on the wounded backs of nude rats, the anastomosis of new lymphatic capillaries from skin grafts and rats’ native lymphatic plexus was observed with lymphatic drainage away from the wounds. Of note, the study highlighted the use of fibroblasts in engineering lymphatic capillaries as in the absence of fibroblasts, hLECs did not develop into capillaries, suggesting that fibroblasts are required to provide a suitable physiological environment for microvessel formation.

Other sources of hydrogel include hyaluronic acid (HA), which is one of the components of natural extracellular matrices and known to play a significant role in wound healing [89]. In lymphatic tissue engineering, HA was used in a rat myocardial infarction model to effectively decrease collagen deposition, recover cardiac function, and induce VEGF-positive vasculature formation [90]. The mix of hyaluronic acid and methylcellulose (HAMC) loaded with VEGF-C and Ang-2 was used to treat a sheep hindlimb lymphedema model [96]. The growth factors were released by diffusion from the HAMC hydrogels, improved lymphatic function and reduced lymphedema. Matrigel is also utilized for tissue engineering [97,107]. Matrigel is an ECM protein mixture derived from mouse Englebreth–Holm–Swarm tumors and composed of laminin, collagen IV, and enactin. One of the important applications of Matrigel is for the growth of stem cells. Therefore, it can be useful when stem cells are used for lymphatic engineering. When Matrigel with MAPCs was transplanted into a mouse wound, it supported lymphatic capillary growth, regeneration of lymphatic network and drainage [97]. Another study used Matrigel to deliver MSCs and EPCs in mice, which demonstrated robust formation and development of lymphatic vessels although functionality of these vessels were not studied [107]. Despite its widespread use, Matrigel is not a well-defined matrix and therefore can introduce variability in experimental results [91].

In order to promote lymphangiogenesis, growth factors, especially VEGF-C have been used for local in vivo delivery. Growth factors can be immobilized to a biopolymer component of hydrogels and released locally, mimicking endogenously released growth factors [92]. In vitro lymphangiogenesis was observed using fibrin hydrogels with covalently bound VEGF and interstitial flow [93]. The study tested various compositions of fibrin and collagen and found that the fibrin-only matrix was the most suitable for LEC organization and capillary formation. The use of VEGF-C within hydrogels was further supported by Guc et al., in which fibrin-binding VEGF-C (FB-VEGF-C) mixed with fibrin hydrogels was used in a mouse ear skin and diabetic skin wound healing model and observed to stimulate local lymphangiogenesis in a dose-dependent manner [92]. Furthermore, the study confirmed the physiologic and immunologic functions of newly formed lymphatic capillaries. This approach was also used to induce therapeutic lymphangiogenesis for cardiac remodeling after cardiac ischemia and reperfusion injury [108]. In this study, a gelatin-based hydrogel was used to deliver VEGF-C to the injured myocardium and shown to increase lymphatic density and attenuate the development of ischemic-induced heart failure.

Hydrogels have also been used to form lymph nodes or mimic the functionality and microenvironment of lymph nodes. One in vitro approach was to devise a bioreactor that mimics the microenvironment of a lymph node and allows immune cell clustering and activation. The bioreactor allows the perfusion of cells and medium into the central culture space. The central culture space contained either agarose hydrogels or nonwoven polyamide fiber sheets which housed human dendritic cells differentiated from monocytes [95]. The two matrices were chosen based on cell attachment, porosity, stability, and cell growth. At the end of 2 weeks, both matrices showed sustainable lymphocyte clusters containing antigen-specific leukocytes. This system can be a useful tool to study drug and cell interaction within physiologic environments. More recently, injectable hydroxypropyl methylcellulose (HPMC) hydrogels were used to harbor human adipose derived stem cells (hADSCs) and regenerate lymph nodes in vivo. In this study, the matrix was supplemented with TGFβ1 and basic fibroblast growth factor (bFGF) [109]. There have been studies to use hydrogels to mimic the lymph node microenvironment for immunological study, such as cancer progression and therapeutics, suggesting its role as a novel tool to study immunologic pathology and therapy [98,110].

Hydrogels are a great scaffold material for tissue engineering. Natural materials that are described here have several advantages for tissue engineering such as biocompatibility, degradability, and intrinsic cellular interaction and function [89]. However, they exhibit less ideal mechanical strength, which warrants further investigation and modifications to improve this limitation.

### 5.2. Non-Biodegradable Synthetic

Another strategy is to use a nanocomposite polymer, the UCL-Nano (polyhedral oligomeric silsesquioxane poly(carbonate-urea) urethane [POSS-PCU]). This polymer has a superior surface nanotopography and viscoelasticity than other synthetic scaffolds [111]. The nanocomposites have advantageous properties compared to conventional materials, such as high surface area-to-mass ratio, mechanical strength, and biostability. In addition, its cytocompatibility shows no significant difference in cell viability, adhesion, and proliferation compared to standard cell culture plates, supporting its suitability for tissue engineering [99]. This polymer has shown promising cardiovascular applications such as heart valves and bypass grafts [111].

### 5.3. Biodegradable Synthetic

Biodegradable scaffolds provide a temporary scaffold that can be substituted with cellular matrix and avoid material-related side effects. Polyglycolic acid (PGA) is one of the commonly used biodegradable scaffolds with well-documented biocompatibility and ability to custom design [100]. A preliminary work by Dai et al. used the PGA scaffold and LECs as a seed cell source to reconstruct lymphatic vessels. They reported a successful LEC endothelialization in vitro and in vivo [100]. Synthetic scaffold offers well-controlled and reproducible environments for cells and growth factors. Their composition and properties can be modified to mimic specific chemical and mechanical properties of cellular environments. However, the scope of achieving desired properties can be limited by technical challenges that will require further innovations.

### 5.4. Biocompatible Natural (Fibrin, Collagen)

Although synthetic scaffolds have demonstrated a potential to be used for tissue engineering, they did not achieve widespread acceptance for clinical uses. Instead, using biocompatible scaffolds with seed cells has gained more popularity.

Fibrin and collagen are natural materials used for scaffolds. They are readily accessible and contain a spectrum of biochemical and physical cues needed for cell morphogenesis and function [9]. They were previously described to promote successful vasculogenesis. Helm et al. composed matrices with varying degree of fibrin and collagen [93]. BECs and LECs were subjected to these scaffolds, matrix-bound VEGF, and slow interstitial flow and in vitro capillary morphogenesis was observed. It was found that the most optimal condition of LEC organization was in a fibrin-only matrix. This study shows that the composition of scaffolds influences cellular behaviors as well as contributions from VEGF and interstitial flow in capillary morphogenesis. Another study supports these findings that fibrin matrix with VEGF-C stimulates local lymphangiogenesis restricted to lymphatic capillaries [92]. When LECs were seeded in fibrin scaffolds with adipose-derived stem cells (ASCs), lymphatic network formation was observed [101]. LECs did not differentiate in the absence of ASCs, implying a positive effect of ASCs on LEC organization. Furthermore, VEGF-C was necessary to induce lymphatic network development. In a porcrine lymphedema model, nanofibrillar collagen scaffolds were implanted across the lymphedema area and a significant increase in functional lymphatic collectors were observed [102]. In another study, LECs and supporting cells, dental pulp stem cells (DPSCs), were cocultured in a collagen scaffold with PDFGR-β signaling and lymphatic vessel formation was observed within a few days [112]. In addition, when mechanical cyclic stretch was applied to the system, the formation of complex muscle tissue aligned with the lymphatic network was observed. When this engineered construct was implanted into a mouse abdominal wall muscle, anastomosis between implant and host lymphatic vasculatures was observed, demonstrating potential functionality.

In the presence of appropriate biochemical stimuli such as VEGF-C and seed cells, natural proteins such as fibrin and collagen can provide adequate structural support for lymphangiogenesis. However, there are several drawbacks: the diversity of biochemical and physical cues may complicate when trying to isolate the interaction of specific cells and growth factors. Furthermore, their predefined chemical and mechanical properties can be frustrating when engineering specific lymphatic tissues.

### 5.5. Cell-Based

Gibot et al. described that human lymphatic endothelial cells (LECs) cocultured with dermal fibroblasts could spontaneously organize into a stable 3D lymphatic capillary network devoid of any exogenous material such as scaffolds or growth factors [103]. In vitro-generated human lymphatic capillaries exhibit the major cellular, ultrastructural and molecular features of native human lymphatic microvasculature, including a lumen, discontinuous basement membrane, blind ends, overlapping junctions, anchoring filaments and branches in 3D. The protocol takes 6 weeks to complete, and it requires experience in cell culture techniques. Matsusaki et al. developed a human skin equivalent (HSE) containing blood and lymph-like capillary networks using a cell accumulation technique [113]. The co-sandwich culture of human umbilical vein endothelial cells and normal human dermal lymphatic microvascular endothelial cells within eight-layered dermis showed in vitro co-network formation of individual blood (CD31^+^ LYVE-1^−^) and lymph-like capillaries (CD31^−^ LYVE-1^+^) inside the dermis. This type of dermis has some great advantages such as precise controllability of the thickness, whole and homogeneous construction, and higher biological functions.

### 5.6. Decellularization/Recellularization

Various studies have shown the potential of organ-derived decellularized extracellular matrix (dECM) scaffolds in engineering organ-like structures. dECM scaffolds overcome the challenge associated with the design and fabrication of synthetic scaffolds. Pre-clinical and clinical application of dECM scaffolds from discarded human donor organs in tissue engineering has been explored by various studies [114,115,116]. Decellularization of a tissue or an organ leaves behind the complex structural compositions and mechanical and functional properties that comprise the extracellular matrix [117]. Combining dECM scaffolds with stem cells or host cells has shown to be a promising tool to reconstruct tissue- and organ-like structures. dECM scaffolds are considered the most similar to the native organ as they retain a unique tissue-specific architecture, provide natural biochemical environment, house biological molecules such as growth factors and cytokines, and avoid immunologic incompatibility and inflammatory responses [104].

Several studies have used the decellularization technique to engineer lymph nodes and lymphatic vessels [105,106,118,119]. Efforts at reconstructing lymph nodes using dECM scaffolds have been carried out by several studies. Cuzzone et al. decellularized lymph nodes from adult mice, repopulated with splenocytes, and implanted in a submuscular pocket of mice. The authors showed a preservation of extracellular matrix architecture of the lymph node and the ability to deliver lymphocytes in vivo [106]. More recently, Lenti et al. reconstructed lympho-organoids using dECM scaffolds and stromal progenitor cells. They transplanted the engineered organoids at the site of resected LNs in mice and observed that the organoids integrate into the native lymphatic vasculature with restored physiologic and immunologic function in vivo [119]. In the study by Yang et al., the authors constructed lymphatic vessels in vitro using lymphatic endothelial like cells that were differentiated from human adipose derived stem cells and decellularized arterial scaffolds [118]. Although the authors did not test the functionality of the engineered lymphatic vessels, the histology of this construct showed that seeded cells proliferated and attached on the surface of the scaffold.

There are challenges with dECM scaffolds including poor cell infiltration and migration due to the dense matrix that remains after decellularization, necessitating excessive times to repopulate the scaffolds. Additionally, other drawbacks of decellularized scaffolds in clinical settings are limited resources and difficulty in surgical manipulation [117]. However, studies have addressed these challenges by customizing dECM matrix to have easier cell seeding and increased porosity with minimal mechanical disruption [120].

### 5.7. 3D Bioprinting

One of the most recent technologies is three-dimensional (3D) bioprinting. Three-dimensional bioprinting produces a complex, high-resolution architecture that recapitulates the physiologic structure and function of human tissues. Living cells and biomaterial containing biochemical or biophysical cues can be positioned precisely into a predefined architecture [104,121]. The use of 3D bioprinting is rapidly expanding, especially in tissue engineering such as blood vessels and vascularized tissues including heart, liver and kidney, and biomedical applications such as in vitro disease model and drug testing [122].

The feasibility of the 3D bioprinting of vascular grafts has been demonstrated [123,124]. Kolesky et al. used multimaterial 3D bioprinting and fabricated a vascularized tissue with extracellular matrix, vasculature, and multiple cell types. They co-printed vascular, cell-laden and silicone inks. Cell-laden inks contained human mesenchymal stem cells, human neonatal dermal fibroblasts, and human umbilical vein endothelial cells and ECM were composed of a gelatin and fibrinogen blend. The engineered tissue was actively perfused with growth factors to differentiate human mesenchymal stem cells toward an osteogenic lineage [123]. This study demonstrates the possibility of constructing a thick tissue with cellular heterogeneity that is capable of long-term survival. Given the similarities in the architecture and hierarchy of the lymphatic system to those of the vascular system, techniques used in blood vessel engineering can be translated to lymphatic tissue engineering.

Challenges remain in the proper selection of biomaterials and bioprinting methods to achieve desired structural, mechanical, and biological properties of the tissue [121]. The lymphatic system is a multi-scale structure from collecting vessels to capillaries. Microvasculature, such as lymphatic capillaries, are below the resolution of conventional extrusion-based 3D printing technologies. To overcome this limitation, laser-based printing technologies offer higher printing resolution and therefore can be used to produce capillary-sized microvessels [122]. Another proposed method is to induce cellular remodeling by adding endothelial and supporting cells with growth factors to a 3D printed structure that can drive the spontaneous formation of microvascular networks [125].

## 6. Mechanical Stimuli

### 6.1. Interstitial Flow

The role of interstitial fluid flow in lymphatic formation and regeneration have been extensively studied [3,126,127,128]. The mechanical forces generated by interstitial flow through lymphatic capillaries and their microenvironment modulate and contribute to lymphatic capillary morphogenesis, ECM remodeling by fibroblasts, as well as cell migration and signaling [3]. The role of interstitial flow in lymphatic regeneration was first studied in 2003, using a mouse tail model of skin regeneration. Interstitial fluid channels were observed to form prior to lymphatic endothelial cell organization. Lymphatic vessels organized around the channels and occurred only in the direction of lymph flow, highlighting the significance of interstitial flow in lymphangiogenesis [128]. Further studies elucidated the cooperative roles of VEGF-C and interstitial flow in lymphangiogenesis. Specifically, VEGF-C was reported to induce lymphatic cell proliferation and migration while interstitial flow promoted lymphatic organization and its functionality [26,129].

Given its role in lymphatic morphogenesis and functionality, interstitial flow has been utilized for lymphatic capillary engineering. Interstitial flow was incorporated into scaffolds of different compositions of fibrin and collagen and was observed to affect cellular behaviors. LECs organized best in fibrin-only matrices, which exhibit higher compliance and lower hydraulic permeability [93]. Although matrix composition and organization were shown to determine the cell-matrix interaction and subsequently impact cellular function and signaling, interstitial flow proved to be an important component in lymphatic morphogenesis.

### 6.2. Extracorporeal Shock Wave Therapy

Extracorporeal shock wave therapy (ESWT) was first introduced into clinical practice for urologic indication, to fragmentize and remove kidney stones. It is clinically used for the treatment of musculoskeletal diseases including adhesive capsulitis, calcific tendinitis, and plantar fasciitis. ESWT produces an acoustic pressure with positive and negative phases. The positive phase produces a direct mechanical force, and negative phase produces cavitation. Various mechanisms of action have been described for ESWT and one of them being its ability to amplify growth factor and protein synthesis for tissue remodeling [130]. Low-energy ESWT produces biologic effects through mechanical forces such as cavitation and shear stress, which is associated with significant tissue effects. The cavitation causes a sudden collapse of bubbles, increasing cell permeability, inducing increased growth factor expression and activating intracellular signaling pathways [131]. ESWT has been studied in several in vivo models including a rabbit ear lymphedema model [132], rat tail and limb lymphedema models [133,134], and mouse lymphedema model [135]. Animal lymphedema models treated with ESWT demonstrated therapeutic lymphangiogenesis with increased VEGF-C and bFGF expression. When ESWT was combined with VEGF-C hydrogel in mice, they showed a synergistic effect in inducing lymphangiogenesis and alleviating secondary lymphedema [135].

The findings of in vivo studies support its clinical application for lymphedema. Indeed, ESWT has been shown to be effective in treating lymphedema. The use of ESWT on the treatment of upper and lower extremity lymphedema was first reported in 2010. Seventy-two geriatric patients with primary or secondary lymphedema with associated fibrosis received ESWT and demonstrated a reduction in limb circumference and skin thickness [136]. In patients with breast cancer-related lymphedema, ESWT effectively reduced limb volume and circumference [137,138]. More recently, a pilot study by Joos et al. reported a reduction in limb volume and circumference and subjective hardness and edema in patients with an end-stage lymphedema after breast cancer treatment [139]. ESWT can be a great modality to supplement lymphatic tissue bioengineering. It is a non-invasive therapy that eliminates complications or adverse effects associated with invasive procedures or anesthesia. Furthermore, it may be suitable for patients with severe lymphedema who require repeated therapies. The human and animal studies indicate that ESWT promotes angiogenesis, reduces inflammation, and decreases adipocytes, which are involved in the pathologic progression of chronic lymphedema.

## 7. Current State of Lymphatic Tissue Bioengineering

Lymphatic tissue bioengineering efforts are currently focused on a combination of cell-seeded scaffolds with pro-lymphangiogenic factors and external mechanical stimuli to stimulate the growth of lymphatic vessels (Figure 2). Natural and synthetic scaffolds are a delivery system in which cells and pro-lymphangiogenic factors can be delivered to an area of interest and interact with their microenvironment. More recently, decellularization of an existing tissue and 3D bioprinting have emerged as promising strategies to construct a scaffold of desired organization and composition.

Various stem cells have demonstrated potential for lymphatic regeneration including ESCs, iPSCs, HSCs, MAPCs, ECFCs, and ADSCs. Once a reliable cell source for bioengineering is seeded in the scaffold, biomolecular and mechanical stimuli are necessary to stimulate these cells to form lymphatic network. Most widely used pro-lymphangiogenic factor for lymphatic engineering is VEGF-C. Studies to date have recognized pro-lymphangiogenic effects of various factors including VEGF-C, PDGF, FGF, TGF-β, and retinoic acid. Whether these factors exert pro-lymphangiogenic effects independently or supplementary to one another are not well understood. Further studies are needed to find an ideal combination of these factors that can augment cell sprouting and support cell maturation and differentiation [1]. Proliferative effects of growth factors are not limited to lymphatic cell population. Studies have demonstrated that cancer cells also respond to the proliferative signal, limiting their clinical application. However, bioengineered scaffolds allow more precisely targeted delivery of growth factors to the injured area and therefore may allow clinical use of these factors.

Several studies have highlighted the importance of supporting cells. Fibroblasts, ADSCs, and DPSCs have been explored as a co-culture model to support LEC growth. Supporting cells provide pro-lymphangiogenic growth factors and direct cell–cell contact that promote LEC growth. Fibroblasts have been most extensively used with LECs, however, several studies reported that fibroblasts form poor lymphatic vessels and take longer to form lymphatic vessels. Previous studies have shown that ADSCs support vasculogenesis and lymphangiogenesis [100,137,138]. They secrete growth factors including VEGF-A, VEGF-DF, FGF-2, hepatocyte growth factor, and angiopoietin-1. Yet, there is a controversy regarding the secretion of VEGF-C and therefore VEGF-C should be supplemented when ADSCs were used [100]. Importantly, lymphatic networks formed from this system were sustainable over several weeks.

## 8. Challenges

Biomedical applications of bioengineered lymphatic tissues include tissue or organ implantation and reconstruction as well as disease modeling in vitro to study the pathophysiology of the disease as well as drug screening and development. Various strategies to engineer lymphatic tissues have been reported with encouraging in vitro and in vivo data. However, lymphatic tissue engineering has been relatively less studied and therefore, numerous challenges remain to be solved, including design, functionality, and long-term application before it can be implemented in a clinical management and treatment of lymphedema.

Most lymphatic engineering efforts to date have been put toward regenerating lymphatic capillaries with less attempts at constructing larger collecting vessels and lymph nodes. Engineering of a larger and more complex lymphatic structure is especially important for clinical application. In secondary lymphedema, edematous areas are typically large. To effectively drain from large areas and decrease swelling, larger vessels are required for drainage [34]. The research of engineering of a lymphangion, a functional unit of lymphatic vessel capable of intrinsic contraction, has been sparse, owing in part to its anatomical complexities that include intraluminal valves, smooth muscle cells, and pericytes [139]. Given similarities of lymphatic vessels to blood vessels, advances in blood vessel engineering can be translated to lymphatic vessel engineering. The structure of the arterial system consists of a monolayer of endothelial cells and multiple layers of smooth muscle cells. Venous system has valvular structures that prevent the backflow of blood and requires less mechanical strength than arteries do, similar to the requirements of lymphatic vessels. Lymphatic vessels are subjected to a lower flow rate and require less mechanical strength, which may make engineering of lymphatic vessels relatively easier. However, less mechanical strength with a smaller diameter may increase the risk of collapsing. Therefore, the appropriate selection of a scaffold material that can accommodate a low flow rate while preventing collapse is important. Although general techniques of blood vessel engineering can be employed, unique characteristics of lymphatic vessels must be considered.

Another challenge is to design an endothelialized conduit, attaching seed cells onto a scaffold and supporting the growth of these cells. Endothelialization is present in the anatomy of a native lymphatic vessel and therefore essential in engineering a successful lymphatic graft [98]. Additionally, it prevents the possibility of coagulation of a lymphatic vessel. A small-diameter vascular conduit exhibits a high incidence of thrombosis with poor long-term patency in the absence of an endothelial cell layer [98,140]. Although a lymphatic vessel does not hold a risk of thrombosis as a blood vessel does, a low flow rate and a high protein content of lymphatic fluid pose a risk of coagulation in engineered lymphatic channels [141]. Surface topography that supports endothelial cell seeding has been extensively studied for vascular grafts, which can be adapted for lymphatic grafts [141].

Bioengineered lymphatic structures need to be evaluated for the phenotype and functionality. To confirm the lymphatic phenotype, its identifying morphologic characteristics, such as the absence of complete basement membrane and mural cell coverage, and the expression of lymphatic specific markers, such as Prox1, Lyve1, and Podoplanin, can be evaluated. In addition, the ability of bioengineered structures to respond to pro- and anti-lymphangiogenic stimuli should be assessed [35]. Lymphatic functionality, physiologic and immunologic functions, should be confirmed for a successful application of engineered lymphatic tissues. Its function can be assessed by the presence of lumen and its ability of fluid uptake and drainage. Since the analysis of the functionality in vitro can be difficult, different in vivo models can be used. Studies have tested the functionality by implanting engineered tissues into healthy and diseased sites such as lymphedema and wounds. After incorporation of an engineered lymphatic construct in vivo, anastomosis with host lymphatic vasculature and lymphatic drainage is observed using Evans blue dye [94,102].

Recent advances in lymphatic tissue bioengineering have demonstrated potential for its clinical application for lymphedema. Yet, the challenges of designing larger lymphatic vessels within a complex lymphatic system hierarchy remain. With technological innovations and growing research to develop larger-sized and more complex bioengineered tissues and organs, current challenges need to be addressed in future studies.

## Figures and Tables

**Figure 1 bioengineering-09-00162-f001:**
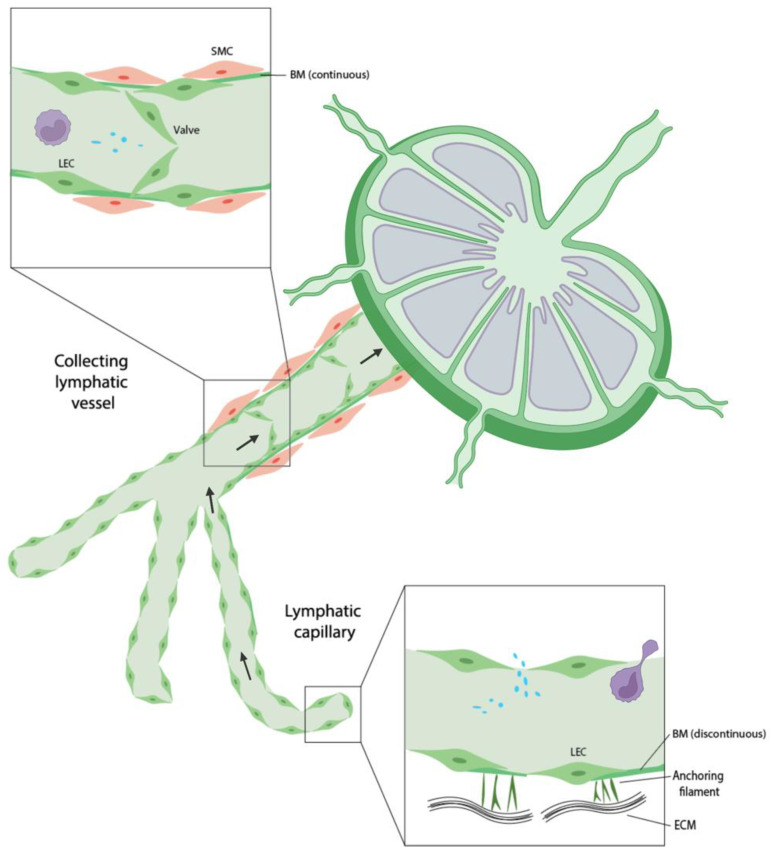
Schematic representation of lymphatic vessels. Lymphatic capillaries and collecting lymphatic vessels possess distinct characteristics.

**Figure 2 bioengineering-09-00162-f002:**
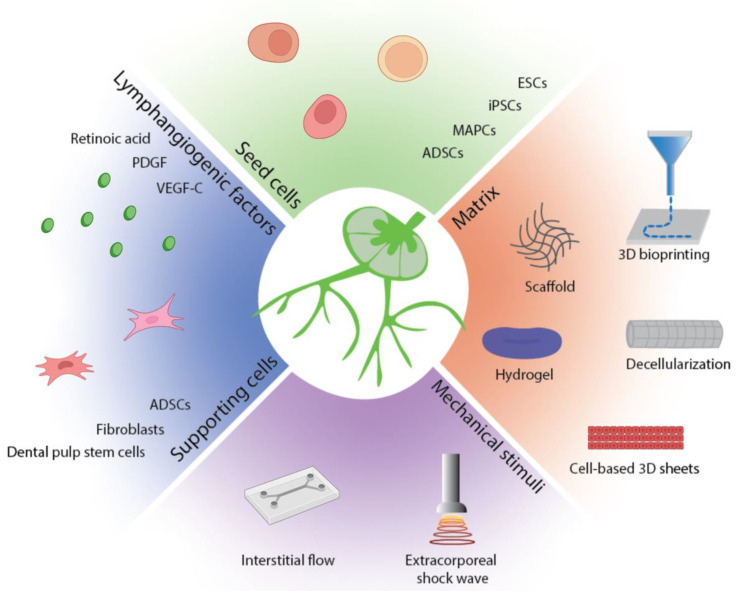
Summary of lymphatic tissue bioengineering strategies. Functional lymphatic tissues can be engineered with seed cells and supporting cells, pro-lymphangiogenic factors, and mechanical stimuli in an appropriate matrix.

**Table 1 bioengineering-09-00162-t001:** Summary of 3D scaffold materials for lymphatic tissue engineering.

Scaffold Material	Model	Structure	Technique/Result	References
Fibrin/collagen hydrogel	In vitro, In vivo	Lymphatic capillaries	hDMECs co-cultured with fibroblasts in fibrin or collagen type I hydrogels to construct skin grafts.	[90]
			Formation of functional lymphatic capillaries observed and growth of lymphatic capillaries and restoration of lymphatic drainage when transplanted to rats.	
	In vivo	Lymphatic capillaries	VEGF-C loaded fibrin hydrogels in mouse subcutaneous cartilage replacement wound healing and diabetic wound healing models.	[91]
			Increased local lymphangiogenesis with immunologic and physiologic functions.	
	In vitro	Lymphatic capillaries	BECs and LECs cultured in different compositions of fibrin and collagen hydrogels with matrix-bound VEGF and slow interstitial flow.	[92]
			Fibrin-only matrix supported LEC organization.	
Gelatin	In vivo	Lymphatic capillaries	VEGF-C loaded gelatin hydrogels in mouse ischemic heart failure model.	[93]
			Increased lymphatic density and transport and attenuated development of ischemia-induced heart failure.	
Hyaluronic acid	In vivo	Lymphatic capillaries	Delivery of hyaluronic acid-based hydrogels to rat myocardial infarction model.	[94]
			Improved ejection fraction, decreased collagen deposition, and increased novel vasculature formation by VEGF staining.	
Hyaluronan and methylcellulose (HAMC)	In vivo	lymphatic capillaries	HAMC with VEGF-C and ANG-2 in sheep hindlimb lymphedema model.	[90]
			Improved lymphatic function and reduced edema.	
Hydroxypropyl methylcellulose (HPMC)	In vivo	Lymph nodes	Delivery of hADSCs from liposuction cultured in differentiation medium, supplemented with TGFβ1 and bFGF to mice.	[95]
			Formation of lymphoid nodes.	
Matrigel	In vitro, In vivo	Lymphatic capillaries and collector vessel	MAPCs and LECs in Matrigel in mouse wound model.	[96]
			Growth of lymphatic capillary in wounds and restoration of lymphatic drainage.	
	In vivo	Lymphatic capillaries	MSCs and EPCs in Matrigel and injected into mice.	[97]
			Growth of blood vessels and lymphatics. Not able to demonstrate functionality of lymphatics	
Non-biodegradable synthetic scaffold	In vitro	Lymphatic vessels	hDLECs in POSS-PCU scaffolds.	[98]
Biodegradable synthetic scaffold	In vitro, In vivo	Lymphatic vessels	hLECs in PGA scaffolds implanted in mice.	[99]
			Development of tubular structures expressing lymphatic markers.	
Fibrin scaffold	In vitro	Lymphatic capillaries	LECs and BECCs co-cultured with adipose-derived stromal cells (ASCs) and supplemented with VEGF-C in fibrin scaffolds.	[100]
			Development of lymphatic network	
Collagen scaffold	In vivo	Lymphatic collectors	Nanofibrillar collagen scaffolds in porcine lymphedema model.	[101]
			Increase in lymphatic collectors within proximity to the scaffolds.	
	In vitro, In vivo	Lymphatic vessel	LECs and DPSCs with PDFGR-β and mechanical cyclin stretch in vitro and implantation into a mouse abdominal wall muscle.	[102]
			Lymphangiogenesis and formation of anastomosis between host and implant lymphatic vasculatures.	
Fibroblast sheets	In vitro	Lymphatic-like capillaries (skin grafts)	Bioengineered dermis with 8 layers of stacked human umbilical vein ECs, LECs, dermal fibroblast, and keratinocyte.	[103]
			Reconstruction of a full-thickness skin tissue with blood and lymphatic-like capillaries in dermis	
Decellularization	In vivo	Lymph node	Lymph nodes harvested from adult mice decellularized, repopulated with splenocytes, and implanted in submuscular pockets.	[104]
			Observation of preserved extracellular matrix architecture and successful in vivo lymphocyte delivery.	
	In vivo	Lymph node	LN stromal progenitors in decellularized lymph nodes upon transplantation at the site of resected LNs.	[105]
			Integration into the endogenous lymphatic vasculature and restoration of lymphatic drainage and perfusion. Activation of antigen-specific immune responses upon immunization.	
	In vitro	Lymphatic vessel	hADSC-differentiated lymphatic endothelial like cells in decellularized arterial scaffold.	[106]
			Cells proliferated and attached well on the surface layer of the decellularized arterial scaffold.	

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
