# Peer review of "Lymphatic Tissue Bioengineering for the Treatment of Postsurgical Lymphedema"

_bioengineering, 2022, doi:10.3390/bioengineering9040162_

Round 1

Reviewer 1 Report

I suggest to accept an article.

Author Response

Thank you for your review

Reviewer 2 Report

The paper of Sung et al entitled ” Lymphatic Tissue Bioengineering for the Treatment of Postsurgical Lymphedema” is a very interesting and complete description of a tissue engineering approaches that could be used in lymphedema, a condition that affect the life quality of different patients. The authors reviewed the stem cells types that can be used, the lymphangiogenic factors and the different types of scaffolds that could be chosen for this goal. In my opinion, the review presents a huge quantity of information in a very synthetic way and based on an important documentation sustained by over 130 bibliographic titles.

Author Response

Thank you for your review

Reviewer 3 Report

In brief, the work is well-written and organized, and easy to follow. The topic seems interesting for potential readership in the field. However, some minor points should be addressed by the authors before further evaluations as below:

  • A schematic should be added to clarify the histological structure of lymphatic tissues
  • Provide some statistical information on lymphatic tissues complications over the globe or the USA
  • Details should be added to the text. For example, the number of cells was injected in the following study “When adipose-derived regenerative cells (ADRCs) were injected directly into...”
  • The current limitations and possible future of stem cells should be mentioned for the treatment of lymphatic disorders
  • “Fibroblast sheets” cannot be placed to “Non-biodegradable synthetic scaffold” in Table 1.
  • Adding more figures is necessary; especially those were taken from histology and immunohistology observations
